# The relationship between professional quality of life and work environment among nurses in neonate care units

Ahmad Batran[1], Ibrahim Aqtam[2]*, Ahmad Ayed[3], Moath Abu Ejheisheh[1]

1 Faculty of Allied Medical Sciences, Department of Nursing, Palestine Ahliya University, Bethlehem, Palestine, 2 Ibn Sina College for Health Professions, Department of Nursing, Nablus University for Vocational and Technical Education, Nablus, Palestine, 3 Faculty of Nursing, Arab American University, Jenin, Palestine

* ibrahim.aqtam@nu-vte.edu.ps; info@nu-vte.edu.ps

## Abstract

### Introduction

The work environment is a critical determinant of the professional quality of life (Pro-QoL) of Neonatal Intensive Care Unit (NICU) nurses. While compassion satisfaction enhances job satisfaction, burnout and secondary traumatic stress have adverse effects on the well-being of nurses and the quality of care provided to patients. This study explores the relationship between the work environment and ProQoL among NICU nurses working in the West Bank, an area plagued by resource scarcity, political instability, and staffing shortages.

### Methods

A cross-sectional study was conducted among 233 NICU nurses in West Bank hospitals from 9 January 2025 to 27 January 2025. Data were collected using the ProQOL and the Practice Environment Scale of the Nursing Work Index (PES-NWI) and analyzed using descriptive statistics, Pearson correlation, and Cronbach's alpha with SPSS version 23.

### Results

Most nurses reported average compassion satisfaction (94.8%), burnout (91.0%), and secondary traumatic stress (84.1%). The practice environment was moderately favorable, mean 2.7 (SD = 0.3), with strong Collegial Nurse-Physician Relations, mean 2.8 (SD = 0.5), and low Staffing and Resource Adequacy, mean 2.6 (SD = 0.4). The Spearman's correlation analysis revealed that a positive relationship existed between the favorable environment and compassion satisfaction (r = 0.747, p < 0.001), while there was a negative correlation with burnout (r = -0.604, p < 0.001) and secondary traumatic stress (r = -0.151, p = 0.021).

**Data availability statement:** The data underlying this study contain potentially identifying and sensitive patient information and cannot be publicly shared due to ethical and legal restrictions imposed by the Institutional Review Board (IRB) at Palestine Ahliya University (IRB approval number: CAMS/BSN/2/2025). Data access requests may be sent to the Ethics Committee at Palestine Ahliya University, which is responsible for evaluating requests and granting data access where appropriate. Interested researchers can contact the committee via email (info@paluniv.edu.ps) or telephone (+970 2-274-1999). Requests should include a brief description of the research purpose and any relevant ethical approvals. Data will be made available to researchers who meet the criteria for access as determined by the Ethics Committee.

**Funding:** The author(s) received no specific funding for this work.

**Competing interests:** The authors have declared that no competing interests exist.

## Discussion

The results suggest improving staffing and resources, nurse-physician collaboration, and emotional support, which are vital in improving ProQoL in a highly demanding environment.

---

## Introduction

Neonatal intensive care nursing is a demanding profession that requires both emotional and physical resilience due to its high-acuity environment and the complexity of care provided to critically ill newborns [1,2]. Nurses in Neonatal Intensive Care Units (NICUs) provide specialized care, often in high-stress environments that can impact their Professional Quality of Life (ProQoL) [3]. ProQoL consists of three key components: Compassion Satisfaction (CS), Burnout (BO), and Secondary Traumatic Stress (STS) [4]. Compassion Satisfaction refers to the fulfillment nurses experience from their work, whereas Burnout and STS arise from prolonged occupational stress and repeated exposure to patient suffering, respectively [5].

A well-structured work environment enhances job satisfaction and mitigates stress, whereas inadequate staffing, high workloads, and limited emotional support can lead to emotional exhaustion and decreased care quality. The World Health Organization (WHO) underscores that healthcare workers' well-being is vital for maintaining high-quality patient care [6]. Research from high-income settings shows that supportive work environments contribute to higher ProQoL by reducing burnout and improving job satisfaction [7,8]. Conversely, studies in low-resource settings highlight the negative effects of staffing shortages and high workloads on nurses' emotional well-being [3,9]. Despite extensive research on ProQoL globally, little is known about how these factors affect NICU nurses in the West Bank, where political instability and resource limitations present additional challenges.

This study examines the relationship between ProQoL and the work environment among NICU nurses in the West Bank. Using Kanter's Theory of Structural Empowerment, which posits that access to resources, support, and professional opportunities enhances job satisfaction and reduces stress [10], this research seeks to provide evidence-based recommendations to improve the work environment and support NICU nurses' well-being in resource-constrained settings.

### Literature review

Professional Quality of Life (ProQoL) refers to the overall well-being of healthcare professionals as they navigate the rewards and challenges of their work [4]. Stamm conceptualized ProQoL as comprising three key components: Compassion Satisfaction (CS), Burnout (BO), and Secondary Traumatic Stress (STS). Compassion Satisfaction reflects the positive emotional benefits of caregiving, such as fulfillment from helping others [4]. Burnout arises from prolonged occupational stress, leading to emotional exhaustion, cynicism, and reduced professional efficacy. Secondary

Traumatic Stress occurs when healthcare workers experience emotional distress from repeated exposure to patient suffering [2,5].

NICU nurses are particularly vulnerable to burnout and secondary traumatic stress due to the high-acuity nature of neonatal care, emotionally demanding family interactions, and frequent exposure to infant mortality. Studies from high-income countries have emphasized the importance of institutional support mechanisms, such as counseling services and resilience training, in reducing burnout [1,7]. In resource-limited settings, however, research highlights the detrimental effects of workforce shortages, excessive workloads, and limited access to psychological support services, which exacerbate emotional exhaustion among NICU nurses [3,9].

The work environment is a key determinant of ProQoL. According to Kanter's Theory of Structural Empowerment (1977), access to resources, leadership support, and professional opportunities enhances job satisfaction and reduces stress. Empirical studies show that supportive work environments are associated with higher compassion satisfaction and lower burnout among nurses [7,8]. Conversely, poor staffing, inadequate resources, and lack of nurse autonomy are linked to higher levels of burnout and secondary traumatic stress [8,11].

The Practice Environment Scale of the Nursing Work Index (PES-NWI), developed by Lake (2000), is one of the most widely used tools for assessing the nursing work environment. It comprises five key subscales: Nurse Participation in Hospital Affairs, Nursing Foundations for Quality of Care, Nurse Manager Ability, Leadership, and Support, Staffing and Resource Adequacy, and Collegial Nurse-Physician Relations [12].

Research using the PES-NWI suggests that hospitals with favorable practice environments report lower burnout rates and higher job satisfaction [13]. A study by Liu et al. found that nurse-physician collaboration and adequate staffing levels were the strongest predictors of ProQoL [8]. However, in low-resource settings, staffing shortages, high patient loads, and limited leadership support negatively impact nurses' well-being [3,9]. Recent studies in Sub-Saharan Africa (e.g., Kenya) and South Asia (e.g., India) further highlight how systemic resource constraints and sociocultural dynamics exacerbate burnout among NICU nurses [14,15].

NICU nurses working in resource-constrained regions, such as the West Bank, face unique socio-political and economic challenges. Chronic staffing shortages, limited access to medical supplies, and unstable political conditions further burden NICU nurses, exacerbating stress and burnout [6]. Similar trends have been reported in other low-resource settings, where lack of emotional support programs and high patient-to-nurse ratios contribute to severe burnout and emotional exhaustion [14].

Despite the well-documented effects of work environment on nurse well-being, research focusing on NICU nurses in the West Bank remains scarce. Findings from similar resource-limited healthcare settings suggest that targeted interventions; such as workload redistribution, peer support programs, and leadership training, can enhance ProQoL [15,16,17]. This study seeks to bridge this gap by examining the relationship between ProQoL and work environment among NICU nurses in the West Bank, offering insights into potential solutions for improving nurse well-being in under-resourced healthcare settings.

## Methodology

### Design

This study employed a cross-sectional design to examine the correlation between Professional Quality of Life (ProQoL) and the work environment among nurses in Neonatal Intensive Care Units (NICUs) in the West Bank. Data collection took place from January 9, 2025, to January 27, 2025.

### Population and sampling

The target population consisted of over 500 NICU nurses in the West Bank. Using the Raosoft tool, with a 5% margin of error, a 95% confidence interval, and a 50% response rate, the estimated sample size was determined. To account

for potential attrition, the final sample size was set at 260, with 233 nurses ultimately participating, yielding an 89.6% response rate. A stratified random sampling method was used to ensure proportional representation from each hospital. Inclusion criteria required participants to be full-time NICU nurses present during data collection. Part-time nurses and those on rotating shifts were excluded to maintain consistency in ProQoL assessment.

## Instruments

The study utilized a structured questionnaire comprising the following sections:

1.  **Socio-demographic Information:** This section captured variables such as age, gender, education level, work experience, shift type, and sleep hours.

2.  **Practice Environment Scale of the Nursing Work Index (PES-NWI):** Developed by Lake (2000), the PES-NWI is widely used to assess nursing practice environments and identify factors that support or hinder quality patient care. The instrument consists of five subscales: (1) Nurse Participation in Hospital Affairs, (2) Nursing Foundations for Quality of Care, (3) Nurse Manager Ability, Leadership, and Support, (4) Staffing and Resource Adequacy, and (5) Collegial Nurse-Physician Relations [18]. The PES-NWI contains 31 items, with each subscale comprising between 3 and 10 items. Reliability estimates using Cronbach's alpha have been reported between 0.71 and 0.84. The PES-NWI subscales are rated on a 4-point Likert scale ranging from 1 (strongly disagree) to 4 (strongly agree), with higher scores indicating a more supportive and well-structured practice environment.

3.  **Professional Quality of Life Scale (ProQOL) Version 5:** Developed by Stamm (2010), this 30-item scale assesses three subscales: Compassion Satisfaction, Burnout, and Secondary Traumatic Stress [4]. Respondents rate their experiences on a 5-point Likert-type scale ranging from 1 (never) to 5 (very often), with total scores categorized as low, average, or high. The ProQOL has demonstrated high reliability and validity across over 200 peer-reviewed studies, with Cronbach's alpha values of 0.86 for Compassion Satisfaction, 0.88 for Burnout, and 0.84 for Secondary Traumatic Stress in this study.

## Data collection

Data collection occurred over a three-week period. Questionnaires were distributed in English, with Nurse Managers confirming participants' proficiency to minimize language barriers. The research team provided a brief orientation session to explain the study's purpose, survey completion instructions, and confidentiality measures. Participants completed the surveys in a designated quiet area at their workplaces to minimize disruptions. Completed questionnaires were returned in sealed envelopes to maintain anonymity, and responses were coded for confidentiality.

## Data analysis

Data were analyzed using SPSS version 23 (SPSS Inc., 2015). Descriptive statistics summarized participant characteristics. Pearson's correlation assessed relationships between work environment factors and ProQoL components. Multiple linear regression analysis was conducted to predict the impact of work environment factors on ProQoL. Instrument reliability was confirmed using Cronbach's alpha.

## Results

### Participant characteristics

Of the 260 questionnaires distributed, 233 were returned (response rate 89.6%). The majority of respondents were aged 30 years or younger (59.7%), and their sex was female (73.8%). Over half reported having a bachelor's degree (64.8%), while the majority had less than five years of experience (73.0%). A majority of the nurses worked rotating shifts (62.7%), and 50.6% reported sleeping eight hours per night. As seen in Table 1 (see S1 Table in S1 File for detailed demographics)

**Table 1. Demographic characteristics of participants.**

| Characteristic | Category | n (%) |
|---|---|---|
| **Age** | ≤30 years | 139 (59.7) |
| | 31–40 years | 75 (32.2) |
| | >40 years | 19 (8.2) |
| **Gender** | Male | 61 (26.2) |
| | Female | 172 (73.8) |
| **Education Level** | Diploma | 64 (27.5) |
| | Bachelor's | 151 (64.8) |
| | Master's+ | 18 (7.7) |
| **Experience** | ≤5 years | 170 (73.0) |
| | 6–10 years | 36 (15.5) |
| | >10 years | 27 (11.6) |
| **Work Shift** | Day shift | 87 (37.3) |
| | Rotating shift | 146 (62.7) |
| **Sleep Hours** | <8 hours | 59 (25.3) |
| | 8 hours | 118 (50.6) |
| | >8 hours | 56 (24.0) |

## Professional quality of life

The majority of respondents reported average levels of compassion satisfaction, 94.8%; burnout, 91.0%; and secondary traumatic stress, 84.1%. Only a few had high compassion satisfactions, 5.2%, while no participant had high burnout. This suggests that the nurses do get some fulfillment from their job roles but at the same time undergo much emotional strain. As shown in Table 2 (see S2 Table in S1 File for full breakdown)

## Practice environment

The overall practice environment was considered moderately favorable, with a mean of 2.7 (SD = 0.3). Collegial Nurse-Physician Relations had the highest ranking, with a mean score of 2.8 and an SD of 0.5, indicative of strong relationships between professionals; Staffing and Resource Adequacy ranked lowest at 2.6 (SD = 0.4), reflecting critical shortages in key resources. As shown in Table 3 (see S3 Table in S1 File for subscale scores)

## Correlation analysis

Spearman's correlation analysis showed that a favorable practice environment was significantly associated with higher compassion satisfaction (r = 0.747, p < 0.001) and negatively correlated with burnout (r = -0.604, p < 0.001) and secondary traumatic stress (r = -0.151, p = 0.021). Among the PES-NWI subscales, Collegial Nurse-Physician Relations and Nurse Participation in Hospital Affairs exhibited the strongest relationships with compassion satisfaction and burnout. A multiple linear regression analysis was conducted to explore the predictive power of work environment factors on

**Table 2. Professional quality of life levels.**

| Variable | Low (%) | Average (%) | High (%) |
|---|---|---|---|
| **Compassion Satisfaction** | 0 (0.0) | 221 (94.8) | 12 (5.2) |
| **Burnout** | 21 (9.0) | 212 (91.0) | 0 (0.0) |
| **Secondary Traumatic Stress** | 35 (15.0) | 196 (84.1) | 2 (0.9) |

**Table 3. Practice environment scores.**

| Domain | Mean (SD) |
| --- | --- |
| Nurse Participation in Hospital Affairs | 2.7 (0.3) |
| Nursing Foundations for Quality of Care | 2.7 (0.3) |
| Nurse Manager Ability, Leadership, and Support | 2.7 (0.4) |
| Staffing and Resource Adequacy | 2.6 (0.4) |
| Collegial Nurse-Physician Relations | 2.8 (0.5) |
| **Total Practice Environment** | **2.7 (0.3)** |

ProQoL components. The model explained 58% of the variance in compassion satisfaction ($R^2 = 0.58$, $p < 0.001$), 47% in burnout ($R^2 = 0.47$, $p < 0.001$), and 21% in secondary traumatic stress ($R^2 = 0.21$, $p = 0.019$). Staffing and Resource Adequacy was the strongest predictor of burnout ($\beta = -0.49$, $p < 0.001$), while Collegial Nurse-Physician Relations was the most significant predictor of compassion satisfaction ($\beta = 0.52$, $p < 0.001$). These findings suggest that improving staffing levels and professional relationships could enhance ProQoL outcomes. As depicted in Table 4 (see S4 Table in S1 File for regression coefficients).

## Subgroup analysis

Subgroup analyses revealed that female nurses had slightly lower levels of compassion satisfaction compared to their male counterparts, though these differences were not significant ($p > 0.05$). Compassion satisfaction was higher, while burnout was lower for those nurses with over ten years of experience compared to less experienced groups of nurses, which was statistically significant ($p < 0.05$). Also, the practice environment scores were more favorable for those nurses having a master's degree than for those lesser qualified ($p < 0.05$).

These findings indicate the potential points of targeted interventions, including support for inexperienced nurses and those with particular needs linked to their educational background. The present study, therefore, also calls for resolving staffing and resource shortages while capitalizing on strong interpersonal relationships in order to improve ProQoL and quality of neonatal care.

## Discussion

This study explored the relationship between Professional Quality of Life (ProQoL) and the work environment among NICU nurses in the West Bank. The findings indicate that staffing shortages, limited resources, and nurse-physician relationships significantly impact nurses' ProQoL, reinforcing existing research on the subject.

**Table 4. Correlations between practice environment and ProQoL.**

| Variable | Compassion Satisfaction (95% CI) | Burnout (95% CI) | Secondary Traumatic Stress (95% CI) |
| --- | --- | --- | --- |
| **Nurse Participation in Hospital Affairs** | .596 (0.001)** [0.50, 0.68] | -0.515 (0.001)** [-0.61, -0.42] | .126 (0.055) [-0.01, 0.27] |
| **Nursing Foundations for Quality of Care** | .577 (0.001)** [0.48, 0.66] | -0.604 (0.001)** [-0.69, -0.51] | .119 (0.069) [-0.02, 0.26] |
| **Nurse Manager Ability, Leadership, and Support** | .482 (0.001)** [0.38, 0.57] | -0.425 (0.001)** [-0.52, -0.33] | .143 (0.029)* [0.01, 0.28] |
| **Staffing and Resource Adequacy** | .540 (0.001)** [0.44, 0.63] | -0.236 (0.001)** [-0.35, -0.12] | .004 (0.950) [-0.14, 0.15] |
| **Collegial Nurse-Physician Relations** | .683 (0.001)** [0.60, 0.75] | -0.397 (0.001)** [-0.49, -0.29] | .165 (0.012)* [0.04, 0.29] |
| **Total Practice Environment** | .747 (0.001) [0.68, 0.81] | -0.604 (0.001) [-0.69, -0.51] | .151 (0.021) [-0.28, -0.02] |

*Significant at the 0.05 level.

**Significant at the 0.01 level.

Most participants (94.8%) reported moderate Compassion Satisfaction, suggesting that NICU nurses derive fulfillment from their roles despite workplace challenges. However, only 5.2% reported high Compassion Satisfaction, indicating that resource limitations and excessive workloads may hinder greater job satisfaction. Similarly, while 91.0% of nurses reported average Burnout, none reported high Burnout levels, suggesting the presence of coping mechanisms. However, continued exposure to excessive workloads and emotional strain could increase burnout risk over time. Research in high-income settings suggests that workplace wellness programs and emotional resilience training reduce burnout [11,19], yet such interventions are often underdeveloped in resource-limited settings like the West Bank.

The study also found that 84.1% of nurses experienced moderate levels of Secondary Traumatic Stress (STS), with 15.0% reporting low STS. This highlights the emotional burden NICU nurses face due to frequent exposure to neonatal mortality and distressing family interactions. Similar studies have shown that NICU nurses are particularly vulnerable to STS due to the intensity of neonatal care and the emotional demands of supporting grieving families [1]. Given the political instability and healthcare constraints in the West Bank, these challenges may be even more pronounced compared to high-income settings.

The study underscores the significant influence of work environment factors on ProQoL. Collegial Nurse-Physician Relations emerged as the highest-rated factor (mean = 2.8, SD = 0.5) and was positively correlated with Compassion Satisfaction (r = 0.683, p < 0.001). This supports research demonstrating that strong interdisciplinary collaboration enhances job satisfaction and reduces burnout [7,8]. Conversely, Staffing and Resource Adequacy received the lowest rating (mean = 2.6, SD = 0.4) and was negatively correlated with Burnout (r = -0.604, p < 0.001). Previous studies have shown that staffing shortages and inadequate resources contribute to severe burnout and emotional exhaustion, particularly in low-resource settings [14].

Regression analysis revealed that Staffing and Resource Adequacy was the strongest predictor of Burnout (β = -0.49, p < 0.001), while Collegial Nurse-Physician Relations was the strongest predictor of Compassion Satisfaction (β = 0.52, p < 0.001). These findings reinforce Kanter's Theory of Structural Empowerment, which asserts that access to resources, leadership support, and professional collaboration enhances job satisfaction and reduces stress [10].

Addressing these challenges requires targeted interventions. Improving staffing levels and ensuring adequate resources can mitigate burnout and enhance patient care quality. Strengthening nurse-physician collaboration through structured training programs and shared decision-making protocols can further improve job satisfaction. Additionally, introducing workplace wellness programs, peer support networks, and mental health resources can help nurses manage STS and burnout more effectively.

Overall, this study highlights the critical need for systemic improvements in NICU work environments to enhance Pro-QoL. Future research should focus on longitudinal studies to assess changes over time and qualitative investigations to explore nurses' lived experiences in greater depth. Intervention-based research evaluating the effectiveness of workplace wellness programs and resilience training could further inform policies aimed at improving NICU nurses' well-being in resource-limited healthcare settings.

## Implications for practice and policy

Addressing staffing and resource shortages involves increasing nurse-to-patient ratios, as implementing higher staffing levels could mitigate burnout and improve patient care quality. Ensuring that NICU nurses have access to adequate medical supplies is also critical for reducing workplace stress. Enhancing interdisciplinary collaboration should focus on improving nurse-physician communication through training programs aimed at strengthening interdisciplinary teamwork, as positive nurse-physician relations were a strong predictor of ProQoL. Additionally, establishing shared decision-making protocols can empower nurses in clinical settings. Developing emotional support programs is essential, starting with the implementation of structured resilience training. Studies indicate that workplace wellness programs, which reduce burnout in high-income settings, could be adapted for resource-limited hospitals to enhance NICU nurses' emotional well-being.

Providing peer support networks and dedicated psychological services can help nurses manage secondary traumatic stress more effectively. Finally, investing in professional development by offering continuous stress management training, through regular workshops on mental health coping strategies, will help nurses navigate high-pressure environments. Leadership training for nurse managers will also be essential to promote a supportive workplace culture.

### Study limitations

While this study provides valuable insights, several limitations should be acknowledged. First, the cross-sectional design of the study does not establish causal relationships between work environment factors and ProQoL. Future research should consider using longitudinal designs to assess changes over time. Additionally, the use of self-reported data introduces the potential for social desirability bias, as nurses might have underreported their burnout levels. Combining survey data with qualitative interviews could provide richer insights. The findings also have limited generalizability, as they are specific to the West Bank and may not fully apply to other healthcare systems. Comparative studies with other resource-limited regions could enhance understanding. Lastly, cultural factors such as societal expectations of resilience or stigma around mental health in Palestinian society [14,15] may influence nurses' willingness to report stress or burnout. For instance, cultural norms emphasizing stoicism could lead to underreporting of emotional strain, skewing ProQoL assessments.

### Future research directions

Future research should include longitudinal studies to track changes in ProQoL over time, allowing for a better understanding of how work environment factors influence nurses' well-being in the long term. Qualitative research, such as interviews, could further explore nurses' lived experiences and coping mechanisms, providing deeper insights into the emotional challenges they face. Comparative studies in other resource-limited healthcare settings would help identify universal versus region-specific determinants of ProQoL, enhancing the generalizability of findings. Additionally, intervention studies should be conducted to test the effectiveness of workplace wellness programs and resilience training, determining their impact on improving nurses' ProQoL in various healthcare environments.

## Conclusion

This study provides critical insights into the relationship between Professional Quality of Life (ProQoL) and the work environment among Neonatal Intensive Care Unit (NICU) nurses in the West Bank. To translate findings into action, policymakers should advocate for enforceable nurse-patient ratio laws, and allocate funding for workplace wellness programs. Hospital administrators could adopt WHO's guidelines on staffing benchmarks and integrate peer support networks to address cultural stigma around mental health.

## Supporting information

**S1 File. Tables (S1–S5) provide detailed demographic data, professional quality of life scores, work environment metrics, correlation analysis, and statistical methods.**
(DOCX)

## Acknowledgments

The authors would like to express their thanks to the nurses who participated in the study

## Author contributions

**Conceptualization:** Ibrahim Aqtam, Ahmad Ayed, Ahmad Batran.

**Data curation:** Ibrahim Aqtam, Ahmad Ayed.

**Formal analysis:** Ibrahim Aqtam, Moath Abu Ejheisheh, Ahmad Batran.

**Investigation:** Ahmad Ayed, Moath Abu Ejheisheh, Ahmad Batran.

**Methodology:** Ibrahim Aqtam, Ahmad Ayed, Moath Abu Ejheisheh.

**Project administration:** Ibrahim Aqtam.

**Supervision:** Moath Abu Ejheisheh.

**Writing – original draft:** Ibrahim Aqtam, Ahmad Ayed.

**Writing – review & editing:** Ibrahim Aqtam, Moath Abu Ejheisheh, Ahmad Batran.

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
