## [Decision Letter · Decision Letter 0]

5 Mar 2025

PONE-D-25-05013The Relationship between Professional Quality of Life and Work Environment among Nurses in Neonate Care UnitsPLOS ONE

Dear Dr. aqtam,

Thank you for submitting your manuscript to PLOS ONE. After careful consideration, we feel that it has merit but does not fully meet PLOS ONE’s publication criteria as it currently stands. Therefore, we invite you to submit a revised version of the manuscript that addresses the points raised during the review process.

We look forward to receiving your revised manuscript.

Kind regards,

Amal Diab Ghanem Atalla, ph.D

Academic Editor

PLOS ONE

Additional Editor Comments:

Dear Author,

Thank you for submitting your manuscript to PLOS ONE Journal. After careful evaluation by our reviewers and the editorial team, we have determined that your manuscript has merit but requires substantial revisions before it can be considered for publication.

The reviewers have identified several key areas that need significant improvement, including [briefly mention the main concerns, e.g., methodological limitations, insufficient data analysis, unclear presentation of results, or inadequate discussion]. We encourage you to carefully address each of the reviewers' comments and provide a detailed response outlining the changes made. If any suggestions cannot be implemented, please provide a clear justification.

Given the extent of the required revisions, your manuscript will undergo another round of peer review upon resubmission. Please ensure that your revised manuscript adheres to the journal’s formatting and reporting guidelines.

We appreciate your efforts and look forward to receiving your revised manuscript. Please do not hesitate to reach out if you require any clarification.

Best regards,

Amal Diab Ghanem Atalla

Academic editor at PLOS ONE Journal

Reviewers' comments:

Reviewer's Responses to Questions

**Comments to the Author**

1. Is the manuscript technically sound, and do the data support the conclusions?

Reviewer #1: Yes

Reviewer #2: Yes

2. Has the statistical analysis been performed appropriately and rigorously? 

Reviewer #1: Yes

Reviewer #2: Yes

3. Have the authors made all data underlying the findings in their manuscript fully available?

Reviewer #1: Yes

Reviewer #2: Yes

4. Is the manuscript presented in an intelligible fashion and written in standard English?

Reviewer #1: Yes

Reviewer #2: Yes

5. Review Comments to the Author

Reviewer #1: Areas for Improvement

Clarity and Conciseness

Some sections, particularly the Introduction and Discussion, could be more concise.

Certain sentences are repetitive, especially in explaining concepts like burnout and secondary traumatic stress. Consider streamlining these explanations.

Depth of Literature Review

While the study references relevant literature, it could benefit from more recent global studies on NICU nurses’ well-being.

The discussion would be strengthened by incorporating additional sources that examine how similar challenges are addressed in other resource-limited settings.

Statistical Interpretation

The Spearman’s correlation analysis is appropriately used, but a regression analysis could provide more insight into predictive relationships between work environment factors and ProQoL components.

Consider adding confidence intervals for correlation coefficients to provide more statistical clarity.

Limitations Section

The study acknowledges limitations (cross-sectional design, self-reported data, geographic constraints), but it could be expanded by discussing potential response bias or the impact of cultural factors on ProQoL perceptions.

A recommendation for qualitative research (e.g., interviews with NICU nurses) could further enrich future studies.

Conclusion and Future Directions

The conclusion effectively summarizes the findings but could be stronger by emphasizing specific next steps for research or policy implementation.

Future studies could explore interventions (e.g., workplace wellness programs) to improve ProQoL outcomes rather than just measuring relationships.

Reviewer #2: The Relationship between Professional Quality of Life and Work Environment among Nurses in Neonate Care Units

Dear authors, thank you for trying to deepen the knowledge within your topic, I have taken great interest in reading your work.

- Please ensure that your manuscript meets all requirements

- The title is very critical and important, additionally it covers an extra point of research not covered in previous research.

_ In the abstract you have mention the conclusion, the discussion you have removed it

-In Literature review, the first paragraph; you mentioned directly the dimension ProQOL but at first put the definition and not write also the abbreviation only in the first line without write the complete name of this abbreviation.

- Introduction; Please mention international studies showed the relationship between the two variables to support your introduction.

- In Methodology, “Practice Environment Scale of the Nursing Work Index (PES-NWI): The PES-NWI was developed by Lake in 2000 ( in this instrument please mention the subdimensions names, additionally where the original reference of the tool specifically where Lake (2000) article)

- Professional Quality of Life Scale (ProQOL) Version 5: The following scale contains 30 items and is represented through three subscales ( in this part please mention the name of author who developed this tool)

- In ethical approval part please mention IRB number

- Write more about data collection processes as how do it and also the duration of data collection?

I wish you successfully continued work with your manuscript!

6. PLOS authors have the option to publish the peer review history of their article (what does this mean? ). If published, this will include your full peer review and any attached files.

**Do you want your identity to be public for this peer review?** For information about this choice, including consent withdrawal, please see our Privacy Policy .

Reviewer #1: No

Reviewer #2: No

---

## [Author Response · Author response to Decision Letter 1]

12 Mar 2025

We sincerely thank both reviewers for their insightful and constructive comments. Your feedback has been instrumental in refining our manuscript, and we believe these revisions have significantly improved the clarity, rigor, and impact of our study.

We hope that the changes we have made satisfactorily address your concerns. Please let us know if there are any further refinements needed.

---

## [Editor Report · Decision Letter 1]

16 Mar 2025

The Relationship between Professional Quality of Life and Work Environment among Nurses in Neonate Care Units

PONE-D-25-05013R1

Dear Dr. aqtam,

We’re pleased to inform you that your manuscript has been judged scientifically suitable for publication and will be formally accepted for publication once it meets all outstanding technical requirements.

Kind regards,

Amal Diab Ghanem Atalla, ph.D

Academic Editor

PLOS ONE

Additional Editor Comments (optional):

Dear Author,

I am pleased to inform you that your manuscript

has been accepted for publication in PLOS ONE Journal following a rigorous peer-review process.

Your research makes a valuable contribution to the field, and we appreciate your dedication and scholarly efforts.

Our editorial team will now proceed with the final production stages, including formatting and proofreading. You will receive proofs for review in the coming weeks. Please ensure timely communication to facilitate a smooth publication process.

We congratulate you on this achievement and look forward to sharing your work with the academic community.

Best regards,

[Amal Diab Ghanem Atalla]

[Academic editor]
---

## [Editor Report · Acceptance letter]

PONE-D-25-05013R1

PLOS ONE

Dear Dr. aqtam,

I'm pleased to inform you that your manuscript has been deemed suitable for publication in PLOS ONE. Congratulations! Your manuscript is now being handed over to our production team.

Kind regards,

on behalf of

Professor Amal Diab Ghanem Atalla

Academic Editor

PLOS ONE